# Subthalamic nucleus gamma activity increases not only during movement but also during movement inhibition

Petra Fischer[1,2]*, Alek Pogosyan[1,2], Damian M Herz[1,2], Binith Cheeran[2], Alexander L Green[2], James Fitzgerald[2], Tipu Z Aziz[2], Jonathan Hyam[3], Simon Little[3], Thomas Foltynie[3], Patricia Limousin[3], Ludvic Zrinzo[3], Peter Brown[1,2], Huiling Tan[1,2]

[1]Medical Research Council Brain Network Dynamics Unit at the University of Oxford, Oxford, United Kingdom; [2]Nuffield Department of Clinical Neurosciences, John Radcliffe Hospital, University of Oxford, Oxford, United Kingdom; [3]Unit of Functional Neurosurgery, Sobell Department of Motor Neuroscience and Movement Disorders, University College London Institute of Neurology, London, United Kingdom

**Abstract** Gamma activity in the subthalamic nucleus (STN) is widely viewed as a pro-kinetic rhythm. Here we test the hypothesis that rather than being specifically linked to movement execution, gamma activity reflects dynamic processing in this nucleus. We investigated the role of gamma during fast stopping and recorded scalp electroencephalogram and local field potentials from deep brain stimulation electrodes in 9 Parkinson's disease patients. Patients interrupted finger tapping (paced by a metronome) in response to a stop-signal sound, which was timed such that successful stopping would occur only in ~50% of all trials. STN gamma (60–90 Hz) increased most strongly when the tap was successfully stopped, whereas phase-based connectivity between the contralateral STN and motor cortex decreased. Beta or theta power seemed less directly related to stopping. In summary, STN gamma activity may support flexible motor control as it did not only increase during movement execution but also during rapid action-stopping.

*For correspondence: petra.fischer@ndcn.ox.ac.uk

## Introduction

Previous studies have described a neuronal stopping network involving prefrontal and supplementary motor cortical regions, as well as the subthalamic nucleus (STN) (*Aron et al., 2014*; *Jahanshahi et al., 2015*; *Rae et al., 2015*). The STN is well-positioned to cancel actions as it receives cortical input via the hyperdirect pathway and can inhibit the thalamus and brainstem via the basal ganglia output nuclei as well as the striatum via the globus pallidus externus (GPe) (*Mink, 1996*; *Wei and Wang, 2016*). In spite of recent advances in understanding functional and effective connectivity within the stopping network using fMRI (*Rae et al., 2015*, *2016*; *Xu et al., 2016*), the fast temporal dynamics of population activity accompanying the stopping process are not entirely clear.

When rats attempted to cancel an action, increased STN firing activity was found irrespective of whether cancellation was successful or not (*Schmidt et al., 2013*), but more recently, micro-electrode recordings in the human STN revealed two distinct subpopulations that selectively increased firing rate either during successful response inhibition or during motor execution (*Bastin et al., 2014*; *Benis et al., 2016*). Also in the GPe a subpopulation termed arkypallidal cells, which seem to receive input not only from the striatum but also from the STN (*Nevado-Holgado et al., 2014*), has specifically been linked to action cancellation (*Mallet et al., 2016*). It is unclear, though, how

**eLife digest** Being able to stop walking to allow a car to pass is one example of how terminating a movement midway through can be essential for surviving in an ever-changing world. However, people with Parkinson's disease sometimes struggle to stop performing a repetitive movement. Also, they may find themselves stopping despite having intended to keep moving. This inability to control stopping and starting can play havoc with everyday activities such as walking.

Some people with Parkinson's disease find that their symptoms improve after a treatment called deep brain stimulation. Surgeons lower electrodes into specific regions of the brain and use them to block the abnormal electrical activity that causes problems with movement. One of the main brain regions targeted is an area called the subthalamic nucleus. Whenever people initiate a movement, nerve cells in the subthalamic nucleus start to become activated at the same time. This synchronization generates rhythmic waves of activity in the subthalamic nucleus, which are called gamma waves.

To find out whether gamma waves are also involved in stopping a movement, Fischer et al. measured activity in the subthalamic nucleus of nine patients with Parkinson's disease as they performed a finger tapping exercise. The patients had to tap their finger in time with a metronome, but refrain from tapping whenever they heard a high pitched noise. As expected, a burst of gamma waves accompanied the start of each finger tap. However, Fischer et al. showed that an increase in gamma waves also occurred whenever patients successfully stopped a finger tap midway. Gamma waves may thus help people to interact flexibly with the world around them.

Techniques like deep brain stimulation have the potential to manipulate gamma waves. In order to treat symptoms without causing side effects, we need to work out how to target brain waves that are altered in patients, without disrupting other processes. A key step towards achieving this is to understand how brain waves change during essential behaviours such as stopping an on-going movement.

different populations within the basal ganglia are activated in a selective and flexible way. Oscillations, particularly in the gamma band (>30 Hz), have been proposed to be a key mechanism for coordinating spatially separate but functionally related assemblies (*Bosman et al., 2012*; *Fries, 2015*; *Nikolić et al., 2013*; *Schoffelen et al., 2005*). We hypothesized that gamma activity may thus also facilitate coordinated activation of task-relevant subpopulations for efficient movement cancelation. A local field potential study in Parkinson's disease patients, however, has shown increased 55–75 Hz gamma activity when patients failed to stop (*Alegre et al., 2013*), which is in line with the prevailing view that gamma activity is pro-kinetic (*Cassidy et al., 2002*; *Fogelson et al., 2005*; *Litvak et al., 2012*) or related to response vigour (*Jenkinson et al., 2013*). Beta activity, instead, is widely viewed as a marker of broad motor suppression within the STN (*Wessel et al., 2016a*) as well as cortex (*Swann et al., 2012*). High STN beta activity for example was linked to elongated response times during incongruent trials in a Stroop task (*Brittain et al., 2012*) and to stronger suppression of cortico-spinal excitability during speech inhibition (*Wessel et al., 2016a*). However, as movements are known to coincide with decreasing beta and increasing gamma activity (*Joundi et al., 2012*; *Lalo et al., 2008*), comparisons between executed and withheld movements might reflect the lack of movement rather than the stopping process per se.

Ideally, stopping would be recorded as a continuous variable that measures how fast an ongoing movement is terminated instead of whether an action has been started at all. Motor inhibition has traditionally been investigated with stop signal or Go/NoGo tasks, in which movements are triggered by cues (*Huster et al., 2013*; *Swick et al., 2011*). In the stop signal paradigm, subjects press a button in response to a go cue and in some trials a stop signal instructs them to withhold the movement. Go/NoGo tasks instead rely on a large fraction of go trials to catch participants out on rare trials, in which a NoGo cue signals them to withhold the pre-potent motor response. Both tasks require participants to decide whether to stay or move but not to interrupt an ongoing action. Successful stopping is achieved in these tasks by successfully delaying or canceling action initiation rather than terminating an action that is already ongoing. Our aim was to extend existing studies by

investigating rhythmic movements that can be interrupted halfway and are not directly preceded by go-cues but are self-initiated. Patients were asked to tap rhythmically to a metronome. Under these circumstances, subjects anticipate the metronome instead of reacting only after each sound, and so movements can be considered self-initiated. They were instructed to stop upon hearing a different cue that was timed such that they were able to stop only in approximately half of their attempts (*Figure 1*). The neural response to the stop signal was not intermixed with a foregoing response to a go cue as the last metronome sound was delivered about 700 ms prior to the stop signal.

The delay of the stop signal was set by the experimenter after a training period at the start and then kept constant for the rest of the experiment. It was delivered relative to the tap instead of the metronome sound to keep movement variability to a minimum and to prevent the strategy of delaying the tap relative to the metronome sound. This, in combination with the instruction to synchronize accurately to the metronome, provided trains of self-initiated actions that were well-matched across trials. The task was also well-suited to investigate endogenous fluctuations in readiness to stop. We analyzed STN local field potentials (LFP) and scalp electroencephalography (EEG) activity recorded in this task from nine Parkinson's disease patients, who underwent deep brain stimulation surgery. To differentiate volitional motor inhibition from salience detection, six of them were recorded in an additional control condition with identical auditory cues but different instructions. Their task in this condition was to finish the tapping sequence with two more taps upon hearing the stop signal instead of attempting to stop (*Figure 1*).

## Results

### Behavioural results

The mean stop signal delay time with a 55% ± (SD) 10% successful stopping rate was 707 ± 49 ms (range = 620–760 ms). The mean interval between the preceding tap and the unsuccessfully inhibited tap in trials where stopping failed was 864 ± 36 ms, and was significantly shorter than the 900 ms interval dictated by the metronome (Wilcoxon signed-rank test, p=0.004). In these trials, patients would have still had on average 156 ms to stop.

Movement trajectories preceding successfully or unsuccessfully inhibited stops were overlapping (*Figure 1*, trajectories measured by a pressure sensor and goniometer). Thus any electrophysiological differences in this window are unlikely related to movement differences per se.

Stopping performance was quantified as *movement extent*, which was the extent of downward movement after the stop signal relative to the amplitude of the preceding upward movement. 0% movement extent thus refers to a full stop. 50% describes a movement that was interrupted halfway and 100% would correspond to a full tap, i.e. failed stopping. Correlations between *movement extent* and various properties of the last regular tap were computed for each patient and then subjected to t-tests to assess if the Fisher's z-transformed correlation coefficients significantly differed from zero on the group-level. In 7 of 9 subjects, movement extent correlated with the tap-to-sound offset, which indicates that stopping performance was worse when the foregoing tap was relatively late in a trial corresponding to previous results (*Fischer et al., 2016*). However, none of the tested variables were associated with successful stopping after FDR-correction of the resulting p-values (see *Table 1*).

Previous research suggests that a surprising sound alone already elicits motor slowing of verbal reports (*Wessel and Aron, 2013*). We thus checked if the tap performed after the salient 'stop signal' (which served as 'continue signal' in the control condition) was delayed or slowed down in the control condition when stopping was not even required. The median intertap interval directly preceding the stop signal (median ITI = 893 ms) did not differ significantly from the one directly after the stop signal (median ITI = 889 ms, Wilcoxon signed-rank test p=1.0).

### LFP and EEG power differences following the stop signal

We tested for rapid LFP and EEG power changes between the stop cue and the average timing of the tap when inhibition failed, which was on average 156 ms after the cue and puts a limit on the window within which successful movement inhibition had to occur.

The STN contralateral to the tapping hand responded to the stop signal with a 60–90 Hz gamma power increase when compared to activity from the tap before (*Figure 2.1* shows the reference data

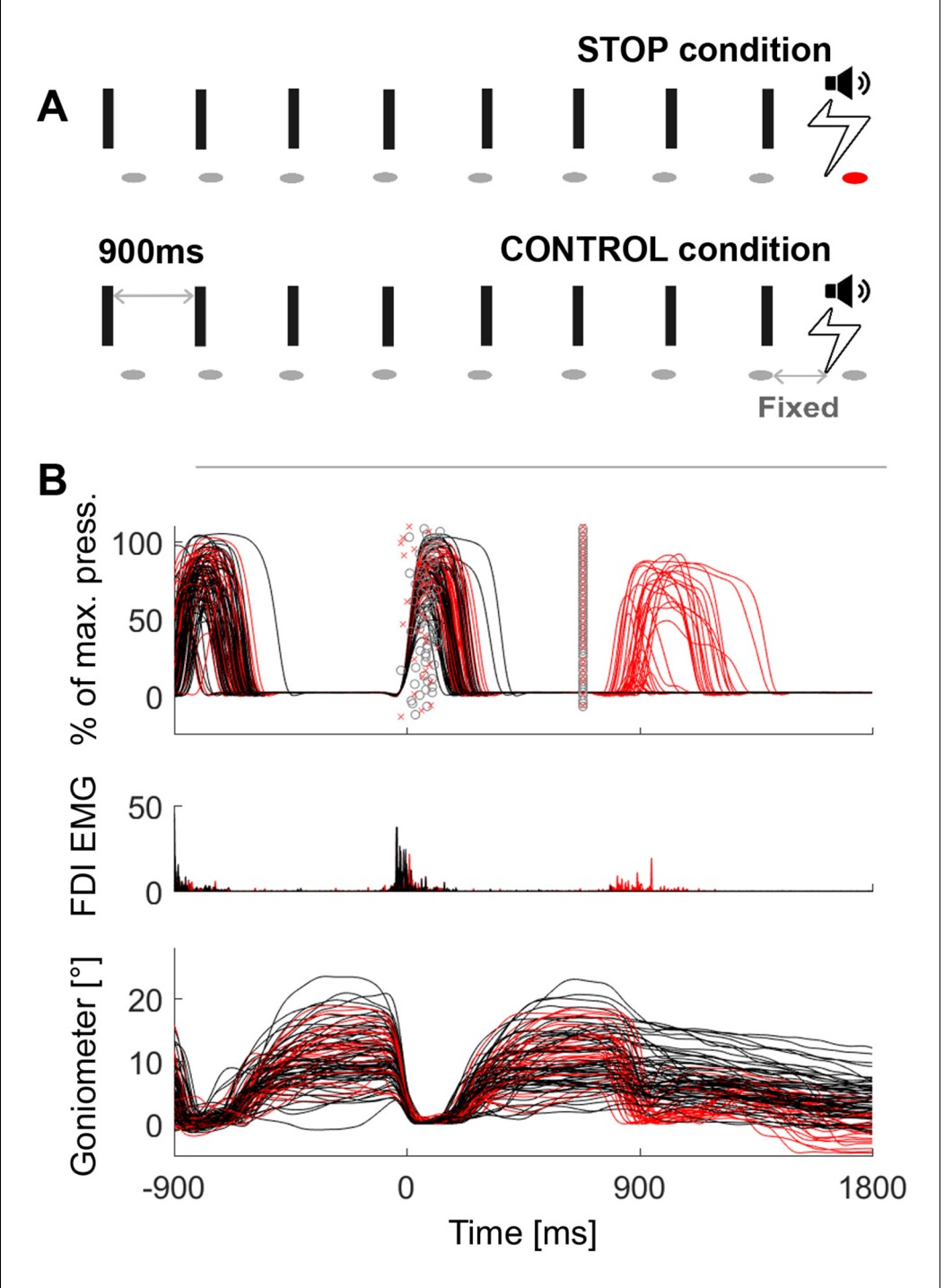

**Figure 1.** Behavioural task and representative data. (**A**) Schematic of the task in the STOP condition (top row) and in the control condition (2nd row). In the STOP condition participants had to tap (=ellipses) to a metronome (=rectangles) and stop after 5–9 taps. The red ellipse denotes a tap that was unsuccessfully stopped. (**B**) Pressure sensor, FDI muscle activity and goniometer data from one representative patient. Black lines are trials where the tapping movement after the stop signal was successfully stopped, red lines are trials where stopping failed. The markers around 0 ms represent the temporal offset between the last regular sound and the tap (o = successful stop trials, x = failed stop trials). The markers at 680 ms show the time of the stop signal, which was always triggered relative to the last regular tap that was registered by the pressure sensor at 0 ms. Note that the black and red trajectories overlap, which shows that stopping performance did not depend on the preceding movement trajectory.

**Table 1.** Correlations between movement parameters of the last regular tap and the movement extent after the stop signal (mean ± SD). In 7 of 9 subjects, movement extent correlated with the soundOffset (=tap-to-sound offset; negative values represent taps that occurred before the sound). But none of the p-values resulting from one-sample t-tests of the Fisher's z-transformed intra-individual correlation coefficients of the nine subjects survived FDR-correction. downTime = duration of finger contact with the pressure sensor, maxPrs = peak pressure during the tap, tapNr = number of taps preceding delivery of the stop signal, peakVelDown=peak velocity of the downward movement of the previous tap, upMvmt = amount of up-movement, peakVelUp=peak velocity of the upward movement.

| Variable | Rho±SD | p-value | FDR-corrected p-value |
| --- | --- | --- | --- |
| soundOffset | 0.29 ± 0.18 | 0.020 | 0.137 |
| downTime | −0.10 ± 0.19 | 0.174 | 0.407 |
| maxPres | −0.04 ± 0.23 | 0.460 | 0.644 |
| tapNr | −0.13 ± 0.20 | 0.061 | 0.215 |
| peakVelDown | −0.05 ± 0.23 | 0.377 | 0.644 |
| upMvmt | 0.00 ± 0.23 | 0.952 | 0.952 |
| peakVelUp | 0.04 ± 0.29 | 0.810 | 0.945 |

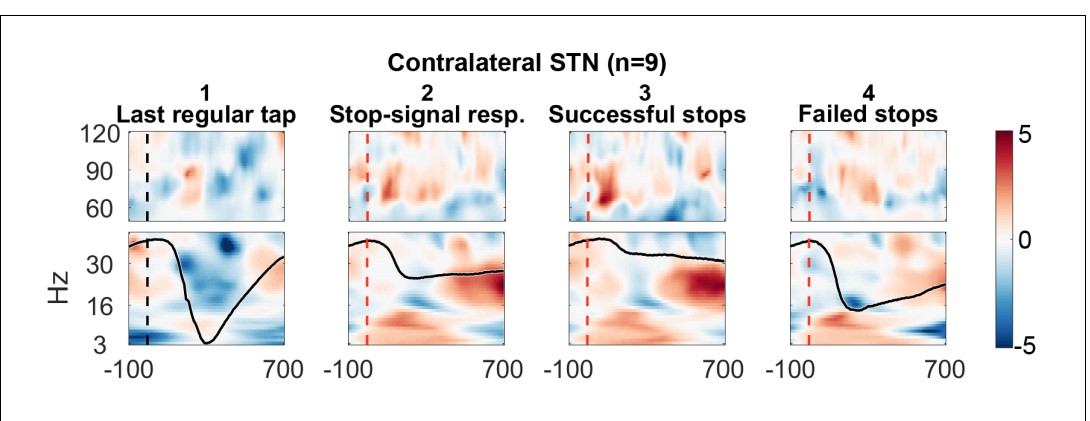

**Figure 2.** Contralateral STN power changes around the stop signal. T-scores calculated over all patients (n = 9, normalized by the average power during regular tapping) for (1) the last regular tap aligned to the timepoint when the stop signal would have occurred if it would have been delivered one tap earlier (vertical dashed line). The black line shows the tapping movement measured by the goniometer. The downward movement was accompanied by a beta decrease and gamma increase as expected. The following three columns show changes in response to the stop signal (vertical dashed line) (2) irrespective of whether stopping was successful or not, (3) during successful stops only, and (4) during failed stops only. Note that when a stop signal was present and especially when stopping was successful (column 3), gamma increased strongly. Differences between 2–1 and 3–4 are contrasted in *Figure 3*. The tapping trajectory of failed stops does not reach the bottom line even though the finger touched the table because trajectories were normalized to the minimum of all four trajectories, which occured with the last regular tap, where the spring was extended more vigorously than during attempted inhibition.

from the tap before aligned to where the stop signal would have occurred if it would have been presented one tap earlier; *Figure 2.2* shows the response to the stop signal; *Figure 3A* shows the contrast between the two). Importantly, this gamma increase was significantly and consistently higher during successful movement inhibition (Figure 2.3 + 2.4, and 3B). The effect size of this difference was very large (60–90 Hz power difference between successful-failed stops: Cohen's d mean$_{winOfInt}$ = 1.2, max$_{winOfInt}$ = 2.6). Note that during regular tapping we observed the typical pattern of movement-related gamma power increase and beta power decrease (*Figure 2.1* and *Figure 3—figure supplement 1*). Gamma power thus increased during both movement execution and movement inhibition. The movement-related peak was broader and weaker than the stop-related increase that peaked around 70 Hz (*Figure 3—figure supplement 2*).

Cortical EEGs recorded a low-frequency increase in response to the stop signal in all channels (*Figure 3A*), which was – in contrast to STN gamma activity – not significantly higher during successful stopping (*Figure 3B*). Only 8–30 Hz power over contralateral C3/C4, Cz and Fz was significantly higher when stopping was successful. However, there was no overall power increase following the stop signal in the 8–30 Hz band in these channels when compared to the tap before (*Figure 3A*), not even when only successful stop trials were considered (*Figure 3—figure supplement 3*). In previous studies, such increase was observed when an action had to be withheld before being initiated (*Kühn et al., 2004*; *Swann et al., 2009*).

To exclude that the STN gamma increase merely reflects processing of the salient stop cue, six patients additionally performed a control condition before and after the main stopping task (*Figure 1*). The stimulus sequence of the control condition was identical to the main condition and the instruction differed only in that patients had to finish the tapping sequence with two more taps upon hearing the stop signal instead of inhibiting the tap immediately. Importantly, no gamma increase was observed in this control condition, even though the difference between successful and unsuccessful stops was still significant despite the reduced sample size of 6 patients (*Figure 4C*).

It has been suggested that specifically the right STN may mediate stopping (*Aron and Poldrack, 2006*). To evaluate the role of the right STN alone, individual gamma differences between successful and unsuccessful stops of all right STNs are displayed in *Figure 4B*, showing no significant increase. Three right-handed patients performed the task with the left hand and thus in those the right STN was the contralateral one. However, in the remaining six, the right STN was the ipsilateral STN, and thus the lack of significant right STN gamma increase indicates that the gamma increase was specific to the contralateral STN.

To further corroborate the functional significance of our finding we also tested whether the average gamma increase peaked earlier during successful stops than during failed stops. Indeed, the average gamma peak of successful stops at 106 ± (SD) 59 ms occurred earlier than the average unsuccessful tap (at 156 ± 50 ms), whereas the average gamma peak of failed stops occurred later (at 179 ± 84 ms). These gamma peak timings significantly differed from each other (t(8)=-2.9, p=0.019, CI$_{diff}$=[−131, −16 ms], Cohen's d = 1.0).

We also examined within-subject correlations between movement extent (i.e. inhibition failure) and gamma within the stopping window (0–156 ms) after the stop signal (see *Figure 4—figure supplement 1*). This was significant in 8 of 9 patients (uncorrected tests; P3's confidence intervals were borderline significant, Spearman's rho p=0.049) when gamma was taken from the contralateral STN, meaning that in all but one patient we found that when gamma was higher, movement extent was less and stopping was more successful. In contralateral C3/C4 and in ipsilateral STN such a relationship was present only in three patients, and in ipsilateral motor cortex only in one patient, further indicating specificity to the correlation with contralateral STN activity. Note though that correlations might be harder to detect with EEG data due to the reduced signal-to-noise ratio in comparison with LFPs.

To see if the gamma increase was highest specifically during full stops, we classified the movement after the stop signal into full stops (<10% downward movement), intermediate stops (>10% but pressure sensor was not touched) and failed stops (all trials where the pressure sensor was touched). Only four patients made five or more full stops (mean number of full stops = 9.5), so formal statistics were not applied. Still, in full stop trials, gamma increased most strongly. It increased moderately for intermediate stops and remained flat for failed stops (*Figure 4—figure supplement 2*). As expected, activity recorded from the first dorsal interosseous muscle of the tapping hand

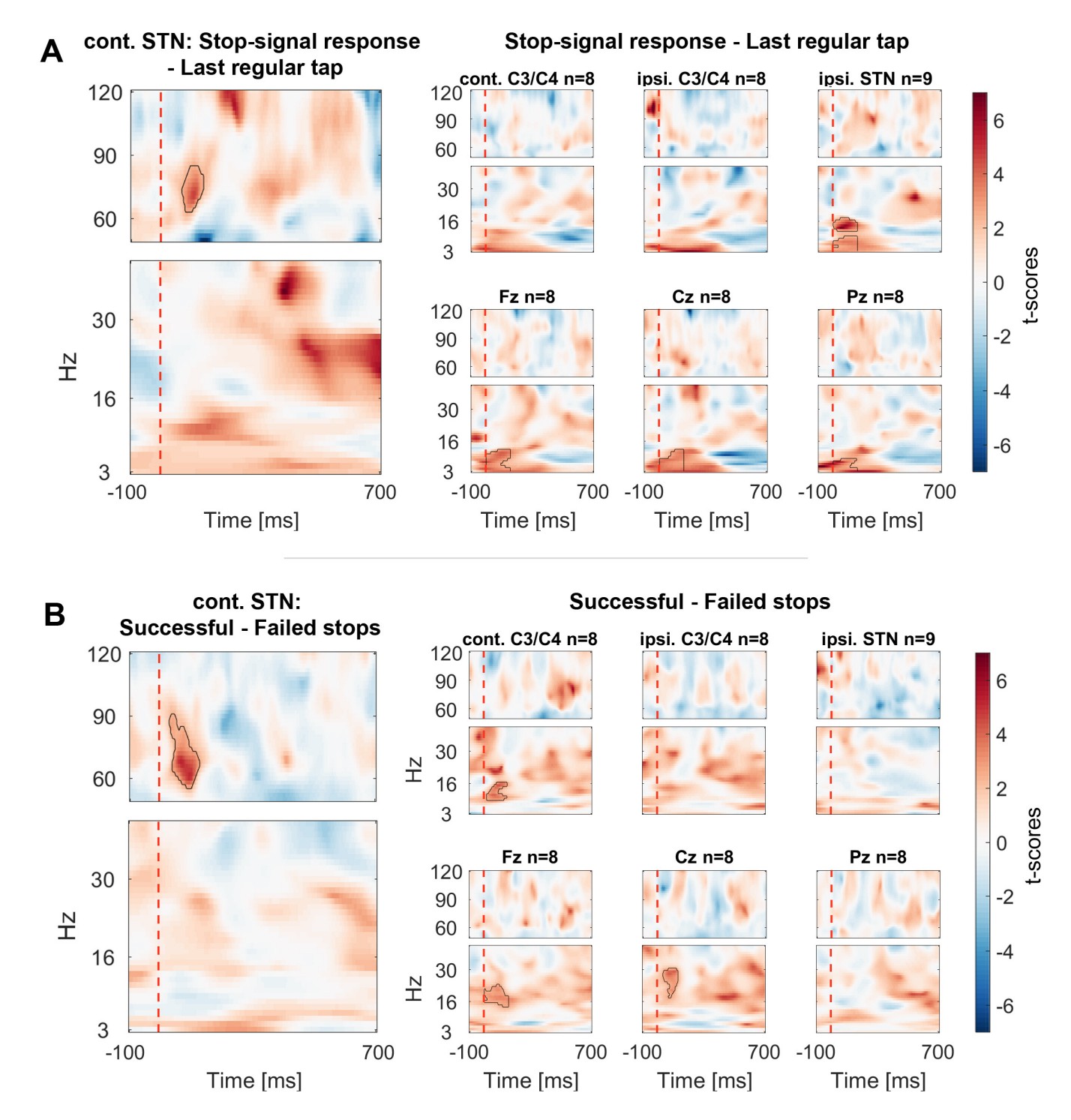

**Figure 3.** Contrasts between power changes following the stop signal. (**A**) T-scores calculated over all patients of the contrast between power aligned to the stop signal (vertical dashed line) averaged across all trials irrespective of stopping performance (*Figure 2.2*) and the regular tap made before (*Figure 2.1*, aligned to where the stop signal would have occurred if it would have been presented one tap earlier). Red clusters denote that power significantly increased in response to the stop signal. (**B**) T-scores of power differences between successful and failed stops. Red clusters denote that power was significantly higher if participants successfully inhibited the upcoming tap (*Figure 2.3–2.4*).

The following source data and figure supplements are available for figure 3:

*Figure 3 continued on next page*

*Figure 3 continued*

**Source data 1.** MATLAB data file containing source data related to *Figure 3*.
Figure supplement 1. Power time-course during regular tapping averaged across all patients.
Figure supplement 2. Peak frequencies of movement- and stop-related power changes.
Figure supplement 3. Power changes following the stop signal when only successful stop trials are considered (averaged across all patients).

(presented to the right in *Figure 4—figure supplement 2*) suggests an inverse relationship to the gamma increase.

Finally, we examined if the cortical 3–5 Hz power increase, which was clearly present in the stop condition (*Figure 3A*), was also present in the control condition when movement inhibition was not even attempted. The grey power trajectory representing the control condition shows a very similar peak in Cz (*Figure 4D*, n = 6). Significance testing within the crucial reaction time window (ranging from the stop signal to the average time of the failed tap, 156 ms later) resulted in no significant differences between the control condition and either the power increase during failed or successful stopping. The direct comparison between failed or successful stops was not significant either. Also a peak-extraction analysis failed to detect a difference between low-frequency peaks (Cz successful stops vs. control: $t(5)=0.2$, $p=0.848$, $CI_{diff}=[-{}-205.1, 240.2\%]$; failed stops vs. control: $t(5)=0.5$, $p=0.641$, $CI_{diff}=[-149.4, 220.8\%]$). The 3–5 Hz increase only seemed to be reduced in the control condition in Fz and both M1 (*Figure 4—figure supplement 3*), however this was also not significant.

## Changes in connectivity between cortex and STN following the stop signal

In a next step, we computed intersite phase clustering (ISPC) values between filtered oscillations in the EEG recordings and the LFP signal from the STN contralateral to the tapping hand. To get an estimate of the temporal development, we subdivided a $-350{:}160$ ms time window around the stop signal into equal bins in which ISPC was computed for each trial and then averaged over trials (see Materials and methods). ISPC describes whether phase differences between two sites are randomly distributed (small ISPC → low connectivity) or clustered (high ISPC → high connectivity) and was obtained by taking the length of the mean vector of all phase differences from all time points within one bin.

ISPC of 60–75 Hz gamma between the contralateral motor cortex and the contralateral STN decreased strongly and significantly in response to the stop signal relative to the average of the $-350{:}0$ ms window preceding the stop signal (*Figure 5*). We also observed an increase of 6–12 Hz ISPC to all cortical channels.

## Power differences preceding the stop signal

Finally we assessed if gamma power was already tonically elevated prior to the stop signal, before participants knew they had to stop. We tested for significant differences within a 350 ms window before the stop signal. If the upcoming tap was inhibited more successfully, STN gamma power was already higher prior to the stop signal (*Figure 6*).

20–30 Hz beta power over C3/C4 ipsilateral to the tapping hand also was significantly higher preceding successful stops. If the data were re-aligned to the last regular tap instead of the stop signal, a second significant cluster at 20–30 Hz over C3/C4 contralateral to the tapping hand was found, in line with previous reports (*Fischer et al., 2016*) (see *Figure 6—figure supplement 1*).

## Discussion

We found that when finger tapping had to be stopped abruptly, the stop signal elicited a fast increase in 60–90 Hz gamma activity in the contralateral STN and a pronounced theta increase in cortex. However, only the former was significantly higher when stopping was successful. The gamma increase occurred within 156 ms, which was the brief time window between the stop signal and the

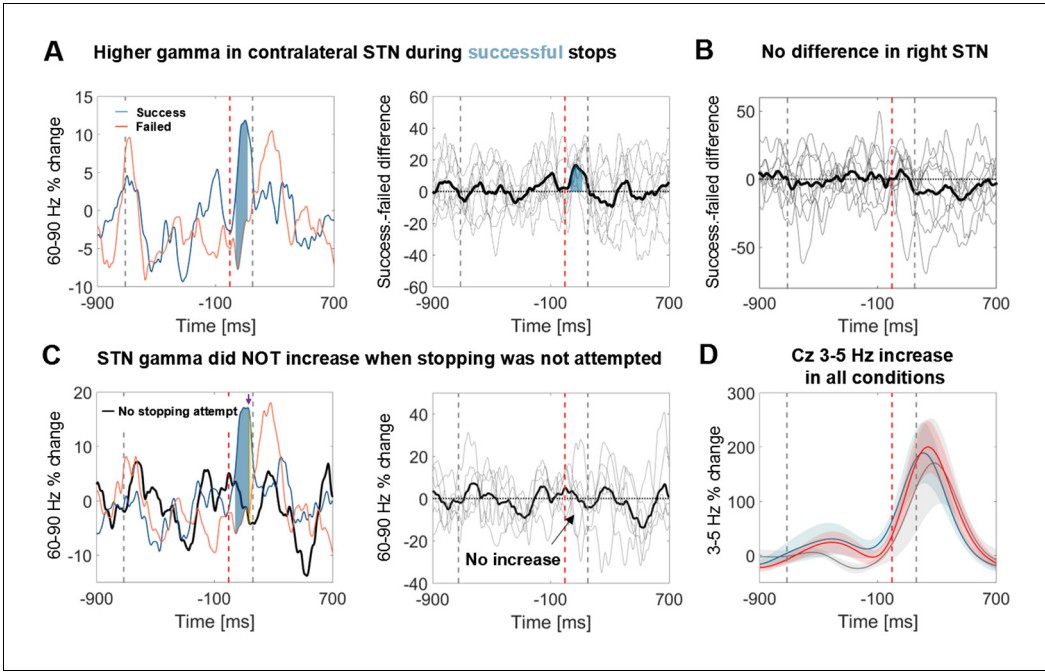

**Figure 4.** Power time course in the STN averaged across patients relative to the stop signal. (**A**) 60–90 Hz gamma power was significantly higher when stopping was successful (left, blue line). The first grey dashed line denotes the average time of the last regular tap. The grey dashed line after the stop signal (red dashed line) denotes the average time of all failed taps. This difference was consistent across patients (middle panel; bold black line denotes the average difference between successful and failed trials with the individual differencess in grey; n = 9). Filled blue areas show cluster-based corrected significant differences. (**B**) This difference was not present in the right STN (n = 9; ipsilateral in 6). (**C**) Gamma in contralateral STN did not increase when stopping was not attempted (black line = control condition, the plot in the middle column shows individual power time courses in the control condition; n = 6). Filled blue areas show cluster-based corrected significant differences between successful and unsuccessful stopping. The yellow filled area indicated by the purple arrow in the leftmost plot shows where power from successful stopping significantly differed from the control condition if uncorrected for multiple comparisons. (**D**) The 3–5 Hz increase in Cz (n = 6) was similar irrespective of whether stopping was successful (blue), unsuccessful (red) or whether it was not even attempted (grey). Shaded areas denote standard errors of the mean.

The following source data and figure supplements are available for figure 4:

**Source data 1.** MATLAB data file containing source data related to *Figure 4*.

**Figure supplement 1.** Scatter plot of correlations between movement extent (x-axis) and 60–90 Hz gamma relative to baseline (y-axis).

**Figure supplement 2.** Power time course relative to the stop signal in patients who stopped fully in at least five trials.

**Figure supplement 3.** 3–5 Hz power increase in contralateral and ipsilateral M1, Fz and Pz.

average failed tap. In a control condition, in which participants were presented with the same stop signal while tapping, but stopping was not attempted, only cortical theta but not STN gamma power increased. This shows that STN gamma activity does not only reflect pro-kinetic activity as previously suggested (*Litvak et al., 2012*) nor does it merely reflect processing of the salient stop signal.

The alternative hypothesis that stopping of the tapping movement itself involved an active movement seems unlikely on two grounds. First, the gamma increase was less if the tap was terminated mid-flight rather than before the downward finger movement was started. Second, gamma connectivity between the STN and C3/C4 also sharply decreased directly after the stop cue, which differs

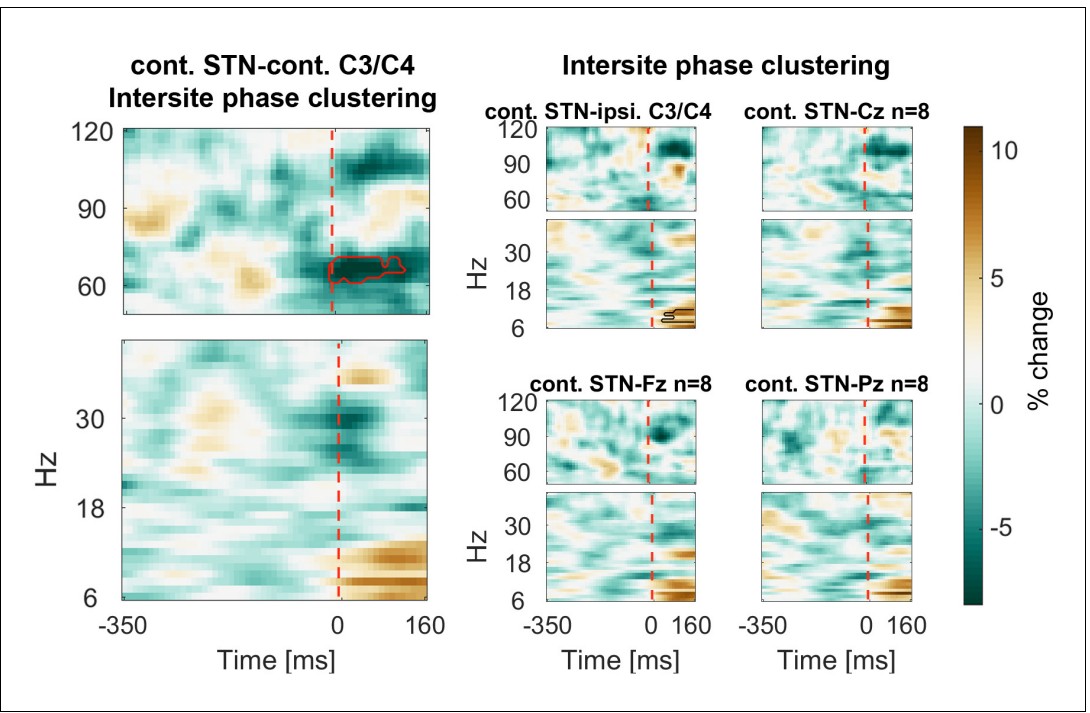

**Figure 5.** Connectivity changes following the stop signal. Intersite phase clustering (ISPC) values are normalized by a −350:0 ms baseline preceding the stop signal. The dashed line denotes the time of the stop signal. Gamma ISPC between contralateral STN and contralateral C3/C4 decreased significantly between 60–80 Hz (encircled in red), whereas ISPC in low frequencies between STN and cortical electrodes increased.

The following source data is available for figure 5:

**Source data 1.** MATLAB data file containing source data related to *Figure 5*.

from the movement-related increase usually observed (*Litvak et al., 2012*) and may indicate disengagement from the obsolete motor plan.

Two previous studies have reported a different relationship between gamma and stopping success to the one that we have found (*Alegre et al., 2013*; *Ray et al., 2012*). Ray and colleagues (2012) reported a gamma increase in response to the stop signal as we do (see *Ray et al., 2012*: Figure 4b) but did not detect significantly higher gamma during successful stops. This discrepancy may result from extensive temporal smoothing (their sliding window was 333 ms long), and from the preselection of a window of interest between 200–400 ms after the stop signal, which also would have failed to detect the gamma difference in our data occurring right after the cue. If we apply the same temporal smoothing to our data, the power trajectories of successful and failed stops would look very similar (data not shown) as extensive smoothing flattens the brief gamma increase, such that part of it appears before the stop signal. The late gamma increase during failed stopping, which would be too late to affect the stopping outcome, would remain as a prominent difference about 300 ms after the stop signal. In Alegre et al.'s study, which differed methodologically in using a visual stop signal, one conclusion was that successful inhibition was associated with a bilateral gamma power decrease. The precise time-frequency decomposition parameters used in that study are unclear and so we have not re-analysed our data in the same way. However, as their window of interest was relatively long (0–0.4 s), they might have predominantly captured the pro-kinetic gamma component that is relatively reduced when a motor response is withheld. Even so, similar as in our study, gamma appeared to increase briefly when aligned to the stop signal in patients on medication during both successful and failed stopping attempts (Alegre: Figure 5) and a drop in STN-M1 coherence during successful inhibition also was found (Alegre: Figure 7). In addition, when comparing results between studies it is important to acknowledge that differences in disease phenotype, medication history, electrode models and in the precision of the targeting achieved are factors that may

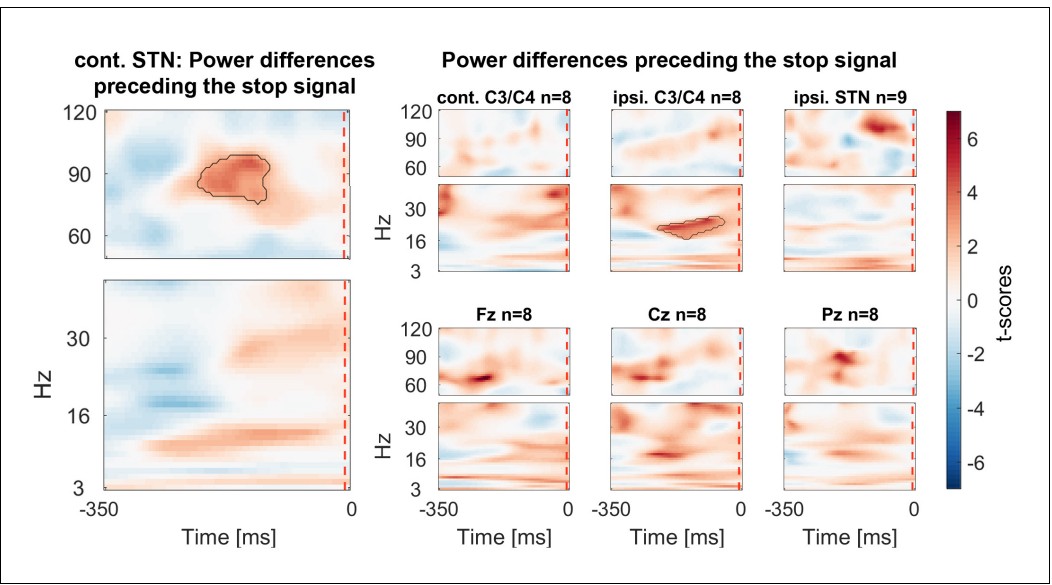

**Figure 6.** Power differences preceding the stop signal averaged across all patients. Around 150 ms before the stop signal (at 0 ms) gamma activity was significantly higher in the STN if stopping was successful. Beta power in ipsilateral C3/C4 was also increased prior to successful stops.

The following source data and figure supplement are available for figure 6:

**Source data 1.** MATLAB data file containing source data related to *Figure 6*.

**Figure supplement 1.** Power differences preceding the stop signal with the data aligned to the last regular tap before stop signal delivery (averaged across all patients).

also contribute to variability in study findings. The normalisation of LFP measures will have only partially mitigated this variability.

Periods of high gamma activity in the STN have been reported to coincide with an overall increase in firing rate and phase-locking of spikes to the gamma cycle peak (*Pogosyan et al., 2006*; *Trottenberg et al., 2006*). From the LFP we cannot infer changes in firing rate, but it suggests that the number of neurons or inputs to these neurons synchronizing at 60–80 Hz was coupled with stopping outcome, and that increased synchronisation occurred early enough to influence such outcome. After observing that the strength of gamma synchronization in the STN or its coherence with C3/C4 did not depend on the exact movement performed, Litvak and colleagues suggested that STN gamma activity modulates rather than explicitly encodes motor commands (*Litvak et al., 2012*). Our results take this hypothesis further by extending the concept of modulation to include a possible role for movement cancelation. This notion is also compatible with observations that have linked STN gamma activity to effort (*Jenkinson et al., 2013*; *Oswal et al., 2013*; *Tan et al., 2013*) and arousal (*Brücke et al., 2013*; *Jenkinson et al., 2013*; *Kempf et al., 2009*). The fact that stopping was more likely successful after STN gamma was relatively high already 200 ms before the stop cue (i.e. before patients knew they had to stop) may reflect such arousal-related function and the need for proactive inhibition.

The present study is correlative in nature, so we cannot infer that gamma oscillations are causally involved in stopping. However, we would like to speculate that a strong surge in STN gamma activity may shift the excitable period of the otherwise observed pro-kinetic gamma increase such that presynaptic spikes arrive at a period of relative inhibition and motor output thus may be interrupted. Inter-individual variability of the peak frequency and the strength of the STN gamma increase may have been related to differences in disease progression, individual stopping speed or electrode placement and type. We did not find a significant gamma increase in cortical electrodes, which may be due to the reduced signal-to-noise ratio of the EEG. However, a broad gamma increase was

observed during stopping in electrocorticography recordings from the pre-supplementary motor area and right inferior frontal gyrus (*Swann et al., 2012*), raising the possibility of cortical involvement in generating the gamma increase via the hyperdirect pathway.

In comparison to tetrode recordings in rats (*Schmidt et al., 2013*), our study is limited in that the recording contacts may not have been directly in the STN. The SNr is located in close proximity, ventrally adjacent to the STN, and thus we cannot exclude that we picked up activity from neighbouring structures. However, gamma has been reported to be specifically localized in the dorsal part of the STN (*Trottenberg et al., 2006*), so that contacts selected according to the strongest gamma modulation are likely located closer to the dorsal border of this nucleus. However, this remains speculative.

Another limitation of this study is that we recorded from patients that may exhibit pathological STN hyperactivity (*Hamani et al., 2004*) expressed in abnormal firing rates and patterns (*Magnin et al., 2000*; *Remple et al., 2011*). Even though these pathological changes are attenuated by dopaminergic medication (*Brown et al., 2001*; *Heimer et al., 2006*; *Levy et al., 2002*), which was taken as usual, and patients were able to perform the task well, neuronal dynamics may still have differed from those of healthy subjects with intact basal ganglia circuits.

It may also be argued that stopping may have involved muscle contractions, which were not picked up by the FDI EMG. But the short latency of the gamma increase and the absence of a similar increase in motor cortex in combination with the decrease in connectivity between STN and C3/C4, which would be expected to increase during movements (*Litvak et al., 2012*), renders this possibility unlikely. Additionally, we observed that gamma increased most strongly in trials where participants were able to stop fully instead of interrupting the downward movement halfway, showing that gamma increased not only during braking in the middle of a movement but that it increased even more in the absence of any movement.

As reported previously, we confirmed a link between higher post-movement C3/C4 beta activity and subsequently improved stopping performance, which we suggest was related to fluctuations in cognitive load (*Fischer et al., 2016*). Beta has also been implicated in time estimation (*Kononowicz and van Rijn, 2015*), thus it may also reflect an intention to delay the next tap's timing, which would allow for more time to stop. Stopping success indeed was correlated with the tap-to-sound offset of the last regular tap in seven patients, such that relatively early taps (early with respect to the sound, which should be compensated for by delaying the next tap) were followed by higher stopping success. Note that this beta differencewas not present in the STN. We also observed significantly higher beta power over contralateral motor and frontal cortex when stopping was successful in comparison to when it failed. As beta oscillations are less likely to occur during movement execution (*Feingold et al., 2015*; *Kilavik et al., 2013*), this difference was expected. In the past, a number of studies have suggested that beta plays an active role in motor inhibition (*Bastin et al., 2014*; *Brittain et al., 2012*; *Wessel et al., 2016a*). Importantly, in the present study no beta increase was observed after the stop signal in comparison to the previous regular tap – not even when only successful stop trials were considered. Thus, it seems unlikely that bursts of beta oscillations per se implemented active braking in our task. Increased beta in other studies may have reinforced the resting position as current motor state that had to be maintained (*Gilbertson et al., 2005*). Such resting posture was not present in our task given that the stop signal was delivered during ongoing tapping.

How can we reconcile the above with the results reported by *Benis et al. (2014)*, who observed a weaker STN beta decrease during 'proactively inhibited' go-trials ('proactively inhibited' as participants were aware that a stop signal may come after the cue, although it did not appear in these trials) in comparison to go-trials with a cue, which was never followed by a stop signal and thus resulted in faster reaction times? The stronger beta decrease may have been related to a more vigorous response in fast go-trials (*Tan et al., 2013*, *2015*) or reduced response uncertainty (*Tzagarakis et al., 2010*) and thus does not necessarily need to reflect an inhibitory process. A stronger difference in beta decrease between the two trial types was also linked to shorter stop signal reaction times across patients. But this correlation may be mediated by symptom severity, as more severe symptoms could result in less beta reactivity (*Little et al., 2012*), reduced modulation of response vigour and longer stop signal reaction times.

Finally, our results may also be reconciled with those reported by *Wessel et al. (2016a)* if elevated beta activity reflects better connectivity across task-relevant areas (*Gross et al., 2004*) or

reduced cognitive load (particularly for <20 Hz beta) (*Fischer et al., 2016*; *Rouhinen et al., 2013*), it could support motor suppression without actually implementing movement inhibition.

Recently, an influential hypothesis suggesting that motor suppression is implemented by fronto-central low-frequency activity has received further support (*Wessel et al., 2016b*). Even though the authors also observed an STN gamma increase concurrent with response slowing, this increase was associated with the cognitive demands of the verbal working memory task rather than motor inhibition. Similar to classical stop signal tasks, our auditory stop cue also elicited a slow-wave power increase. However, this increase occurred also when stopping was not even attempted. If the slow-wave power increase over Cz would have induced slowing or braking, then the intertap interval in the control condition between the tap before and the tap immediately after the stop signal should have been increased, and this was not the case.

Our data suggest an alternative account, namely that the stop signal-evoked slow-wave response does not directly correspond to movement inhibition but instead registers salient sensory stimuli and alerts stopping-relevant areas, which in turn may trigger the STN gamma increase. The increase in cortico-subthalamic low-frequency connectivity might underpin this sequence, enabling the STN to trigger the stopping process. The event-related low-frequency response would thus be necessary for, but not equivalent to motor suppression per se. In Fz and M1 the average low-frequency response seemed to be diminished in the control condition. We would expect that registration of a salient stop signal and efficiency of the transmission process (in terms of speed or extent of neuronal recruitment) depends on endogenous fluctuations in arousal, attention and cognitive load, which would reconcile the hypothesis of low-frequency power-mediated salient stimuli processing with previous results regarding motor inhibition (*Wessel and Aron, 2014*).

Taken together, our results showed that gamma oscillations in the contralateral STN were linked to successful stopping. This indicates that gamma oscillations in the STN are not simply pro-kinetic, but that they can also increase during movement termination. Though we can only infer an association and not causation from observational recordings, our data suggest that the observed gamma rhythm may underpin a fast stopping mechanism involving the STN. Gamma oscillations therefore seem to support fast changes in processing demands not only in cortical but also in cortico-basal ganglia networks in line with theories of gamma synchrony establishing effective, precise and selective neuronal communication (*Fries, 2015*).

## Materials and methods

### Participants

Ten Parkinson's disease patients (mean disease duration = 8 ± 4 years, mean age = 59 ± 8 years; one left-handed/ambidextrous; two female) were recorded after obtaining informed written consent to take part in this study, which was approved by the local ethics committee (Oxfordshire REC A, 08/H0604/58). One patient had to be excluded from the analysis as they intermittently fell asleep during the testing. All patients underwent bilateral implantation of deep brain stimulation electrodes into the STN two to six days before the recording with the aim to alleviate symptoms through chronic high-frequency deep brain stimulation. Surgeries and recordings were performed either at the University College Hospital in London or the John Radcliffe Hospital in Oxford, UK. For each patient one of the following three macroelectrode models were used: Medtronic 3389 (quadripolar, for P1-4 and 8), Boston Scientific, Vercise, DB-2201 (octopolar, for P6) and Boston Scientific, Vercise directional, DB-2202 (octopolar, directional, for P5, 7 and 9). Clinical details of the patients are given in *Table 2*. Patients were tested on medication to ensure task performance and motor function were as normal as possible, although acknowledging that functional impairments, albeit lessened, still persist in this state.

### Task

Participants were asked to tap to an isochronous metronome (900 ms inter-trial interval, ITI, 700 Hz pitch, 40 ms duration) and to interrupt tapping in response to a high pitched auditory stop cue (2000 Hz pitch, 40 ms duration) after a random number of 5–9 taps. The metronome served as regular cue. However, the taps were not triggered in reaction to the metronome sounds but had to be initiated already before the sound to achieve synchronization that depends on how well the timing

**Table 2.** Clinical details. Age and disease duration are given in years. UPDRS-III: Unified Parkinson's disease rating scale part III. Levodopa equivalent dose was calculated according to *Tomlinson et al. (2010)*.

| ID | Age/Sex/dom. Hand | UPDRS-III OFF/ON levodopa | Disease duration | Main symptom | Levodopa equivalent dose (mg / day) | DBS lead | Surgical centre |
|---|---|---|---|---|---|---|---|
| 1 | 65/f/r | 33/11 | 5 | Tremor/Dyskinesia | 807 mg | Medtronic 3389™ | Oxford |
| 2 | 55/m/r | 49/25 | 10 | Leg dragging + tremor (left side) | 2022 mg | Medtronic 3389™ | Oxford |
| 3 | 66/f/r | 25/14 | 17 | Freezing of gait, balance | 1089 mg | Medtronic 3389™ | London |
| 4 | 50/m/r | 37/17 | 5 | Tremor, Dyskinesia, especially in right foot | 958 mg | Medtronic 3389™ | London |
| 5 | 48/m/left-ambi | 46/18 | 6 | Frequent OFFs | 800 mg | Boston Scientific DB-2202™ | Oxford |
| 6 | 54/m/r | 61/32 | 8 | Motor fluctuations | 455 mg | Boston Scientific DB-2201™ | Oxford |
| 7 | 60/m/r | 37/6 | 6 | Rigidity left side, bradykinesia, dyskinesia | 2084 mg | Boston Scientific DB-2202™ | Oxford |
| 8 | 67/m/r | 31/13 | 3.5 | Bradykinesia, Rigidity | 2173 mg | Medtronic 3389™ | London |
| 9 | 68/m/r | 33/15 | 10 | Motor fluctuations | 1765 mg | Boston Scientific DB-2202™ | Oxford |

of the sound is anticipated. If the movement would be reactive, the tap would always lag behind the sound, which was not the case as evidenced by a negative tap-to-sound offset. Thus, this task is special as the metronome cues are not equivalent to go cues

The sound was generated in Spike2 with a 1401 data acquisition unit (Cambridge Electronic Design, Cambridge, UK), played by Creative Inspire T10 speakers and recorded by the EEG amplifier (TMSi Porti amplifier, TMS International, Netherlands). The timing of the stop signal was adjusted in a training period at the beginning such that patients would be able to stop only in 50–60% of all trials. Importantly, the stop signal was triggered relative to the tap registered by the pressure sensor and not to the sound to prevent patients from delaying their taps relative to the metronome, which would improve stopping performance if the latter were the case. A ~50% success rate was desirable to capture fluctuations in alertness or stopping readiness and to distinguish related brain processes. The actual average stopping probability was 55% ± (SD) 10%. Six patients were additionally recorded in a control condition to assess if stopping-related activity was linked to active motor inhibition or whether it merely reflected registration of the more salient stop tone. In this control condition, patients were asked to end the tapping sequence with two more taps after hearing the high pitched sound instead of stopping immediately. The control condition thus posed much less of a challenge than the main stopping task.

A nearly identical task has previously been studied in young healthy subjects (*Fischer et al., 2016*). It differed from the present patient study only in the metronome interval duration, which was shorter (700 ms instead of 900 ms in the patients) and the number of taps (6–10 taps until the stop signal may appear instead of 5–9 taps in the patients). Intervals were chosen to be longer because stopping proved to be more feasible for patients with longer intervals, and the number of taps was reduced to increase the number of trials obtained in the time-limited recording sessions. We planned to record 100 trials in the stopping condition and 20 trials before and after the main block in the control condition. Due to fatigue and time constraints in some cases less trials were recorded. As three patients (P4, P7, P9) had severe motor symptoms on the right side, they performed the task with their left index finger. The remaining six patients used the right index finger. As we would expect the contralateral hemisphere to be more involved in the tapping, we analysed the data not

separated between left and right motor cortex and STN, but between contra- and ipsilateral C3/C4 and STN.

## Behavioural analysis

Behavioural outliers (such as spurious goniometer deflexions) prior to the stop signal were removed following visual inspection. After further exclusion of arrhythmic taps as defined by taps that deviated more than 300 ms from the metronome sound, an average number of 65 ± (SD) 24 trials remained for further analyses. Goniometer traces and the distribution of tap onsets were strongly overlapping prior to successfully vs. unsuccessfully stopped taps (*Figure 1*). To get a graded measure of stopping performance for correlations, the amount of downward movement measured by the goniometer was quantified as *movement extent*. It was defined as the extent of the downward movement normalized by the amplitude of the upward movement done before. The time between the stop signal and subsequently failed stops was quantified as median across trials for each patient and then averaged over subjects.

## Electrophysiological recordings

Bilateral STN local field potentials and EEG was recorded at a sampling frequency of 2048 Hz. EEG electrodes were placed over (or close to if sutures had to be avoided) Fz, Cz, Pz, Oz, C3 and C4 according to the international 10–20 system. Electrooculogram (EOG) was recorded to remove eye blink artefacts in a subsequent procedure. For one patient, EEG channels could not be recorded because large DC drifts caused amplifier saturation. Tap onsets were registered by a force-sensitive resistor measuring the pressure of the finger on its surface. Finger flexion, i.e. the tapping trajectory, was recorded with a goniometer (TMSi Goniometer F35) attached to the index finger over the metacarpophalangeal joint. To capture muscle activity, electromyogram (EMG) was recorded from the first dorsal interosseous muscle (FDI).

## Data pre-processing

Events for tap and sound onsets were created in Spike 2 (RRID:SCR_000903, Cambridge Electronic Design). After DC component removal (2 s time constant), data were processed further with custom routines in MATLAB (RRID:SCR_001622, v. 2014b, The MathWorks Inc., Natick, Massachusetts). EEG channels were re-referenced to linked earlobes if the latter were recorded (n = 5) or to the average of all EEG channels if not (n = 3). LFP bipolars were computed by subtracting two channels of the same recording electrode (bipolar combinations varied depending on the number of available contacts). Data were down-sampled to 1000 Hz and eye artefacts were removed from the EEG signals by subtracting the filtered EOG (40 Hz low-pass Butterworth filter with a filter order of 6, passed forwards and backwards) after amplitude matching via least-squares optimization (MATLAB function *fminocn*). Power between 3–40 Hz was obtained by filtering the data into 3 Hz wide frequency bands shifted by 1 Hz (Butterworth, filter order = 6, two-pass, using fieldtrip functions *_ft_preproc_lowpassfilter* and *ft_preproc_highpassfilter* [RRID:SCR_004849, *Oostenveld et al., 2011*]) and calculating the power of the Hilbert transform. Power between 50 and 120 Hz was calculated within 10 Hz wide frequency bands in 2 Hz steps. To reduce noise, power subsequently was temporally smoothed with a 100 ms sliding window. Before exporting the data, it was further down-sampled to 200 Hz. MATLAB analyses scripts for this procedure and subsequent steps to reproduce the figures are provided as source code files (Source code 1). The data can be downloaded from the Oxford University Research Archive: https://ora.ox.ac.uk/objects/uuid:54c00c3d-1809-4a52-bba8-b491b6075f35.

## LFP bipolar selection

As we recorded from three different electrode models, with multiple contacts of which some may not have been located in the STN, we decided to pre-select the bipolar configuration that recorded the strongest gamma reactivity during regular tapping. We chose to select the contacts based on gamma activity because gamma has been found to be highly focal to the STN (*Trottenberg et al., 2006*). For the quadripolar (Medtronic 3389) and the unsegmented octopolar model (Boston Scientific DB-2201), bipolars were computed between neighbouring contacts or if channels saturated and thus could not be recorded, the surrounding contacts were instead used for the bipolar subtraction. For the directional contacts (Boston Scientific DB-2202), bipolar combinations were computed

between the small segmented ones (C2–C7), plus C1 and C8 if more than two of these channels were saturated to increase the likelihood of including activity from the presumably focal gamma source. As power was converted into relative power changes with respect to a baseline, normalized power estimates were relatively comparable despite differently sized contact surfaces or distances between contacts, as was the case for the directional electrode model.

For the selection process, we first computed the 60–90 Hz median power over all taps for each of the multiple bipolar pairs on each electrode in a time window spanning twice the tapping interval around each tap. Then the range between the maximum and minimum of the resulting power time course was divided by the average power within this window, providing the amount of movement-related gamma modulation captured by each bipolar configuration. For each recording electrode only the bipolar configuration with the highest modulation was analysed further. Note that these contacts also recorded significant movement-related beta modulation as shown in *Figure 3—figure supplement 1*.

## Intersite phase clustering

Phase-based connectivity between the contralateral STN and the five EEG channels of interest (Fz, C3, C4, Cz, Pz) was computed based on the phase of the Hilbert-transformed filtered signal (bandwidth and frequency shifts as described in Data pre-processing). Intersite phase clustering (ISPC) can be defined over trials or over time. As we did not expect high-frequency oscillations to be phase-locked across trials, we calculated ISPC for each trial over multiple fixed-width windows to get an estimate of changes in ISPC over time. The fixed width was 200 ms for 50–120 Hz and 250 ms for 6–40 Hz. The frequency cut-off was 6 Hz as 250 ms would have included only one and a quarter cycle of a 5 Hz oscillation or even less for lower frequencies.

The window width was chosen to be longer for lower frequencies such that more cycles contributed to the estimate. 250 ms would for example encompass four cycles of a 16 Hz oscillation. ISPC was computed within each of these windows, which were shifted by 10 ms such that the overlapping bins resulted in a smooth image. ISPC was obtained by calculating the length of the average vector of phase (ϕ) differences represented as vectors with length one on a unit circle (*Lachaux et al., 2000*) based on the following equation (n=number of samples, MATLAB code provided):

$$\left| \frac{\sum_{t=1}^{n} e^{i*(STN\varphi_t - EEG\varphi_t)}}{n} \right|$$

The amplitude of the signal thus did not contribute to the ISPC estimate. To assess whether ISPC changed in response to the stop signal, we compared whether it differed significantly from zero after normalizing it by the pre-stop signal period ranging from −350 to 0 ms before the stop cue.

## Statistical testing

It should be noted that we analysed LFPs from electrode contact pairs of different surface areas (according to electrode type) and EEGs that had different references between subjects. Accordingly, we only considered normalised changes in power to mitigate this variability. All statistical analyses were performed in MATLAB. Correlations between stopping performance (quantified as movement extent after the stop signal) and movement parameters or features in the EEG/LFP were calculated as Spearman's rank correlation coefficients with bootstrapped confidence intervals (using the *Spearman* function from the Robust correlation toolbox [*Pernet et al., 2012*]). To test if correlations with movement parameters differed significantly from zero on a group-level, correlation coefficients were Fisher's z transformed for each patient and then subjected to a one-sample t-test (n = 9). The maximum correlation with EEG/LFP gamma power (*Figure 4—figure supplement 1*) was determined for each patient by finding the maximum correlation within 60–90 Hz and 0:156 ms after the stop signal.

Each time-frequency matrix was normalized for each subject and frequency by the average power across all regular taps (excluding tap one and those directly followed by a stop signal) to obtain a relative power percentage change before testing for differences.

Multiple-comparison correction for power or ISPC comparisons in time-frequency or time windows of interest was performed by using a cluster-based permutation procedure (*Maris and Oostenveld, 2007*, MATLAB code provided): The original paired samples were randomly permuted 2000 times such that each pair was maintained but its order of subtraction may have changed to

create a null-hypothesis distribution. For each permutation, the sum of the z-scores within supra-threshold-clusters (pre-cluster threshold: p<0.05) was computed to obtain a distribution of the 2000 largest suprathreshold-cluster values. If the sum of the z-scores within a suprathreshold-cluster of the original difference exceeded the 95th percentile of the permutation distribution, it was considered statistically significant.

Pairwise comparisons for behavioural data or peak timings were performed using t-tests or Wilcoxon signed-rank tests if the normality assumption (assessed by Lilliefors tests) was violated and if multiple comparisons were made, p-values were subjected to false discovery rate (FDR)-correction.

## Acknowledgements

This work was supported by the Medical Research Council (MC_UU_12024/1), and PF was funded by a Clarendon Scholarship and St. John's College Award. DMH received funding from European Union's Horizon 2020 research and innovation programme under the Marie Sklodowska-Curie grant agreement number 655605. SL was funded by clinical research training grants from the Wellcome Trust (105804/Z/14/Z). The Unit of Functional Neurosurgery is supported by the Parkinson Appeal UK, and the Monument Trust. The funding bodies played no role in the study design, in the collection, analysis and interpretation of data, in the writing of the report, and in the decision to submit the article for publication.

## Additional information

### Competing interests

BC: has received travel support and unrestricted educational grants for organising CPD events from Medtronic, St. Jude Medical and Boston Scientific (manufacturers of DBS electrodes). TZA: has performed consultancy for and received speaking fees from Medtronic. SL: has been a participant in a DBS teaching course funded by Medtronic, the manufacturer of the electrodes used in this study. TF, PL, LZ: has received speaking fees and travel support from Medtronic and St. Jude Medical. PB: has received fees and non-financial support from Medtronic and personal fees from Boston Scientific. The other authors declare that no competing interests exist.

### Funding

| Funder | Grant reference number | Author |
| --- | --- | --- |
| Medical Research Council | MC_UU_12024/1 | Peter Brown |
| Horizon 2020 Framework Programme | MSCA 655605 | Damian M Herz |
| Wellcome | 105804/Z/14/Z | Simon Little |
| Clarendon Scholarship and St. John's College Award | | Petra Fischer |
| Parkinson Appeal UK | | Jonathan Hyam<br>Simon Little<br>Thomas Foltynie<br>Patricia Limousin<br>Ludvic Zrinzo |
| Monument Trust UK | | Jonathan Hyam<br>Simon Little<br>Thomas Foltynie<br>Patricia Limousin<br>Ludvic Zrinzo |

The funders had no role in study design, data collection and interpretation, or the decision to submit the work for publication.

### Author contributions

PF, Conceptualization, Data curation, Software, Formal analysis, Investigation, Visualization, Methodology, Writing—original draft, Project administration, Writing—review and editing; AP, Conceptualization, Software, Methodology, Writing—review and editing; DMH, Methodology, Writing—review

and editing, Helped with data recordings; BC, Validation, Writing—review and editing, Helped with patient selection and recruitment. B.C. also assessed anatomical electrode locations; ALG, TZA, Writing—review and editing, Performed DBS surgeries; JF, JH, TF, PL, LZ, Writing—review and editing, Performed DBS surgeries and helped with patient recruitment; SL, Writing—review and editing, Helped with patient recruitment; PB, Conceptualization, Supervision, Funding acquisition, Investigation, Methodology, Writing—review and editing; HT, Conceptualization, Supervision, Investigation, Methodology, Writing—review and editing

### Author ORCIDs
Petra Fischer, http://orcid.org/0000-0001-5585-8977
Alexander L Green, http://orcid.org/0000-0002-7262-7297
Peter Brown, http://orcid.org/0000-0002-5201-3044
Huiling Tan, http://orcid.org/0000-0001-8038-3029

### Ethics
Human subjects: Patients were recorded after obtaining informed written consent to take part in this study, which was approved by the local ethics committee (Oxfordshire REC A, 08/H0604/58).

## Additional files

### Supplementary files
• Source code 1. MATLAB analyses scripts to reproduce the figures.

### Major datasets
The following dataset was generated:

| Author(s) | Year | Dataset title | Dataset URL | Database, license, and accessibility information |
|---|---|---|---|---|
| Petra Fischer | 2017 | Subthalamic nucleus activity during stopping of rhythmic finger tapping | https://ora.ox.ac.uk/objects/uuid:54c00c3d-1809-4a52-bba8-b491b6075f35 | Publicly available at the Oxford University Research Archive (https://ora.ox.ac.uk/) |

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
