## [Decision Letter]

Thank you for submitting your article "Subthalamic nucleus gamma activity both promotes and inhibits movement" for consideration by *eLife*. Your article has been reviewed by three peer reviewers, and the evaluation has been overseen by a Reviewing Editor and Sabine Kastner as the Senior Editor. The following individual involved in review of your submission has agreed to reveal his identity: William Hutchinson (Reviewer #2).

The reviewers have discussed the reviews with one another and the Reviewing Editor has drafted this decision to help you prepare a revised submission.

Summary:

Fischer et. al study STN activity recorded from a cohort of nine patients undergoing DBS surgery. The manuscript is a well written and focused on a question of considerable current interest. The authors state that they found that “sudden stopping of rhythmic movement was associated with a pronounced increase of 60-90 Hz gamma oscillations in the subthalamic nucleus”.

The reviewers and the reviewing editor agree that this study addresses a question of a considerable current interest. There are some promising results. The notion that gamma activity was implicated before in promoting action (pro-kinetic rhythm), and here as a stopping signal, is challenging and thought provoking. However, we raised some questions and some comments that should be addressed before the paper can be considered for publication. These are listed below.

Essential revisions:

1) Strength of the concluding statements: The name of the manuscript, its Abstract and large parts of the manuscript suggest a causal effect of the gamma activity on the movement inhibition. The results, however, establishes only correlation between the movement inhibition and the gamma changes, not supporting the claims. This significantly reduces the impact of the findings. The authors are encouraged to revise their interpretation and presentation accordingly.

2) The results seem to contradict earlier findings regarding gamma activity. While this is addressed to some extent in the Discussion and a supplementary file, it would be interesting to validate the claimed reasons for the differences. Thus, it would be useful to consider various options by analysis and further evaluations. For example: 1) perform an analysis on the current dataset using the methods and time windows presented in the earlier articles to compare the results. 2) Consider/discuss the nature of the gamma signal, since it is not consistent across previous DBS LFP studies as the authors point out, and elusive in microelectrode recordings of LFPs and single units over the extent of STN. Βeta and tremor frequencies are robust and ongoing in these recordings whereas gamma is not. In addition to the transient (~100ms) nature of the signal, it has a broad frequency band with an upper bound of 90Hz but the lower bound is not shown since the graph cuts off at 60Hz. In order to clarify this further I think it would be a useful supplemental figure to show the standard power spectrum over the whole frequency range and then show each of the nine patients as overlays behind a bold line showing the mean. This would be a standard way of showing the nature of the gamma peak.

3) While we agree that the paper is well written, this paper is still complicated, and some of the interpretation may mislead some readers. Some revisions and clarifications may reduce this concern:

A) Three different types of electrodes were used in this study. What was the distance between contacts? Could it affect the results, and how?

B) The contacts were chosen based on the gamma band. This choice may induce a bias by selecting the contact that is responding to the finger tapping by an increase of the gamma band. What would have happened if instead, contacts were chosen based on the global changes in LFP activity during finger tapping?

C) How accurate is the contact location ("likely located closer to the dorsal border…"). The authors posit that LFP dynamics in PD patients under treatment may be broadly similar to those in healthy subjects (Discussion, fifth paragraph). There is numerous evidence showing that it's not the case. this caveat should be mentioned and care should be taken in the interpretation of the results. The authors are encouraged to reconsider/discuss or omit this statement. In general, care should be taken when interpreting data from patients and their application to the activity in the normal brain. The activity in patients, even treated, is abnormal in its firing rate, firing pattern and activity during different tasks. The Discussion suggests that the rate is similar, which may be the case, however, many other firing properties are probably different and thus this caveat should be mentioned and care should be taken in the interpretation of the results.

D) The variability between patients should be addressed and clarified. Regarding the evaluation of the right STN and the data in general the authors should perhaps discuss the fact that 3 patients were right handed but performed the task with the left finger.

4) The organization of the results and especially the figures is not in line with the manuscript's main emphasis. Instead of focusing on the contralateral STN, the majority of the figures contain the cortical and ipsilateral STN results. Thus, forcing important results (and especially in their raw unprocessed form) to appear as supplementary figures or not at all. For example: instead of showing in Figure 2 the total power, successful stops and failed stop within the contralateral STN, allowing the observation of the raw phenomenon we see only a processed subset but in all brain areas. It would be useful to focus on the main topic and move the other parts (such as cortical recordings) to the supplementary.

5) The study focuses on different periods and specific time windows which vary between the Results section and figures. It is unclear whether those results are robust under a more general analysis. For example: Figure 2 presents the range -100 to 700 milliseconds but the significance is checked only for 0 to 150 (based on Figure 2—figure supplement 2?). Would testing significance on a different range change the results? The figure seems to hint so, as changes do not seem to end at 150. This is true for other figures and other ranges. The authors are encouraged to revise the figures and where necessary revise the interpretations accordingly.

6) In at least two locations in the Results section, non-significant findings are mentioned as "…did not survive multiple comparison correction over multiple time points due to the small sample size" and continue with a "…but…". This is not acceptable in statistics; something is either significant or not and cannot be forced into a "maybe" state. It should be avoided and simply stated that those results were not significant.

---

## [Author Response]

*Essential revisions:*

*1) Strength of the concluding statements: The name of the manuscript, its Abstract and large parts of the manuscript suggest a causal effect of the gamma activity on the movement inhibition. The results, however, establishes only correlation between the movement inhibition and the gamma changes, not supporting the claims. This significantly reduces the impact of the findings. The authors are encouraged to revise their interpretation and presentation accordingly.*

Thank you for pointing out that some statements were too bold.

We have changed the title to “Subthalamic nucleus gamma activity increases not only during movement but also during movement inhibition” and the last sentence of the Abstract to “In summary, STN gamma activity may support flexible motor control as it did not only increase during movement execution but also during abrupt action-stopping.”

We have also revised large parts of the Discussion and added:

“The present study is correlative in nature, so we cannot infer that gamma oscillations are causally involved in stopping. However, it has been proposed that gamma-band coherence renders neuronal communication selective and precise (Fries, 2015), which may be related to our findings. […]” – The subsequent discourse on the nature of the gamma increase has been added to address revision point 2) (see below).

In the concluding paragraph, we have also added:

“Though we can only infer an association and not causation from observational recordings, our data suggest that the observed gamma rhythm may underpin a fast stopping mechanism involving the STN.”

*2) The results seem to contradict earlier findings regarding gamma activity. While this is addressed to some extent in the Discussion and a supplementary file, it would be interesting to validate the claimed reasons for the differences. Thus, it would be useful to consider various options by analysis and further evaluations. For example: 1) perform an analysis on the current dataset using the methods and time windows presented in the earlier articles to compare the results.*

Comparing to Figure 4 in Ray (2012), our data is shown in Figure 7 (smoothed with a 333ms sliding window, blue line = successful stops, dashed red line = failed stops).

Author response image 1.**DOI:**
http://dx.doi.org/10.7554/eLife.23947.023

The power time course relative to the stop signal at 0ms looks very similar to the one in Ray, 2012. We did not perform visual inspection of the power spectra but kept the frequency band to 60-90 Hz to show how only the smoothing over 333ms already affects the appearance of the gamma increase. The purple shaded areas indicate the pre-stop signal window (-200:0ms) and the post-stop signal window (200-400ms).

If we perform t-tests on the relative changes of the z-scores, we obtain:

t(8)[9]=1.09, p=0.306, Cohens d=0.54

The positive effect size denotes that the relative *(post-pre)/post* power values were higher during successful stopping, which results from the normalization by the *post-window,* which was higher during failed stopping.

If we would normalize by the pre-window: *(post-pre)/pre,* we would obtain a negative effect size, i.e. higher gamma power in failed than in successful stops:

t(8)[9]=-1.06, p=0.319, Cohens d=-0.51

If we would use the original normalization procedure again but on the raw power (and not the z-scores, we would again obtain a negative effect size):

t(8)[9]=-1.73, p=0.122, Cohens d=-0.54

This shows that both the smoothing, and the normalization procedure used in Ray, 2012 can lead to variable results depending on the precise normalization procedure, which could lead to a detection of higher gamma during failed stopping.

Comparing to Figure 5 in Alegre, 2013, the time-frequency decomposition parameters were not entirely clear from the paper (Gabor transform, but with which width/SD?), and thus we did not create another comparable plot. The power time course in Figure 5 shows a gamma increase during failed stops and only a small increase after 0s for successful stops on levodopa (ON, right panel in Figure 5 Alegre, 2013), even though we would expect this gamma increase to be higher. Possibly, the gamma increase would be stronger if the stop signal would be auditory and not visual or if not only the dorsal contacts were analysed. In line with our results, they also found a decrease in M1-STN gamma coherence during successful stopping (Figure 7 in Alegre, 2013).

*2) Consider/discuss the nature of the gamma signal, since it is not consistent across previous DBS LFP studies as the authors point out, and elusive in microelectrode recordings of LFPs and single units over the extent of STN. Βeta and tremor frequencies are robust and ongoing in these recordings whereas gamma is not. In addition to the transient (~100ms) nature of the signal, it has a broad frequency band with an upper bound of 90Hz but the lower bound is not shown since the graph cuts off at 60Hz. In order to clarify this further I think it would be a useful supplemental figure to show the standard power spectrum over the whole frequency range and then show each of the nine patients as overlays behind a bold line showing the mean. This would be a standard way of showing the nature of the gamma peak.*

We have now added a supplementary figure showing the full-range power spectrum from 3-120 Hz (Figure 2—figure supplement 2): It shows the peak frequencies of the movement-related gamma increase, the stop-signal related gamma increase (relative to the last regular tap), as well as the difference between successful and failed stopping. We can see that movement-related gamma was relatively broadly distributed across patients, whereas the stop-related increase was stronger and largest around 70 Hz. We have also extended all the time-frequency plots to range from 50-120 Hz.

In previous studies, STN gamma activity has been observed in conditions requiring higher effort, and we refer to these studies in the Discussion: “This notion is also compatible with observations that have linked STN gamma activity to effort (Jenkinson et al., 2013; Oswal et al., 2013; Tan et al., 2013) and arousal (Brücke et al., 2013; Jenkinson et al., 2013; Kempf et al., 2009).”

But to discuss the nature of the gamma signal further we have now added the following part:

“[…], it has been proposed that gamma-band coherence renders neuronal communication selective and precise (Fries, 2015), which may be related to our findings. The phase of gamma oscillations seems to modulate reaction times in response to a visual stimulus change as well as the gain of multi-unit activity (Ni et al., 2016). […] However, a broad gamma increase was observed during stopping in electrocorticography recordings from the pre-supplementary motor area and right inferior frontal gyrus (Swann et al., 2012), raising the possibility of cortical involvement in generating the gamma increase via the hyperdirect pathway.”

*3) While we agree that the paper is well written, this paper is still complicated, and some of the interpretation may mislead some readers. Some revisions and clarifications may reduce this concern:*

*A) Three different types of electrodes were used in this study. What was the distance between contacts? Could it affect the results, and how?*

The distance between contacts was 0.5mm in the Medtronic 3389 and the Boston Scientific 2201 model. The distance between the directional contacts (2-7) is less in the omnidirectional Boston DB-2202 model and their contact surfaces also are smaller. However, the use of very high impedance amplifiers will have limited the scale of any systematic differences in voltages recorded. In 3 of 5 cases, contact 1 (the lowest one) showed a larger range of voltage fluctuations than contact 8 (=the uppermost contact), whereas in one contact pair this was nearly identical and in another contact pair contact 8 showed higher fluctuations. Most of the small directional contacts (2-7) showed higher voltage fluctuations, but not all of them.

Importantly, we computed bipolar configurations by subtracting two contacts to capture relatively local activity and then normalized the resulting power by computing the% change relative to the average power during regular tapping. Thus, we investigated task-related changes relative to the oscillatory power observed during regular tapping.

We extended the following sentence:

“As power was converted into relative power changes with respect to a baseline, normalised power estimates were relatively comparable despite differently sized contact surfaces or distances between contacts, as was the case for the directional electrode model.”

*B) The contacts were chosen based on the gamma band. This choice may induce a bias by selecting the contact that is responding to the finger tapping by an increase of the gamma band. What would have happened if instead, contacts were chosen based on the global changes in LFP activity during finger tapping?*

The plot in Figure 8 corresponds to Figure 4 but now includes the bipolar contact combinations that showed the highest modulation in the 8-90 Hz range (instead of 60-90 Hz). The new selection procedure resulted in a different bipolar combination for 3 of 9 patients, yet the stronger gamma increase during successful stops is still significant.

Author response image 2.**DOI:**
http://dx.doi.org/10.7554/eLife.23947.024

*C) How accurate is the contact location ("likely located closer to the dorsal border…").*

We have to acknowledge that this remains speculative as we cannot source the post-operative MR/CT data to perform accurate reconstruction of the contact positions. Hence, we added “However, this remains speculative.”

*The authors posit that LFP dynamics in PD patients under treatment may be broadly similar to those in healthy subjects (Discussion, fifth paragraph). There is numerous evidence showing that it's not the case. this caveat should be mentioned and care should be taken in the interpretation of the results. The authors are encouraged to reconsider/discuss or omit this statement. In general, care should be taken when interpreting data from patients and their application to the activity in the normal brain. The activity in patients, even treated, is abnormal in its firing rate, firing pattern and activity during different tasks. The Discussion suggests that the rate is similar, which may be the case, however, many other firing properties are probably different and thus this caveat should be mentioned and care should be taken in the interpretation of the results.*

We have changed the text as follows:

“Another limitation of this study is that we recorded from patients that may exhibit pathological STN hyperactivity (Hamani et al., 2004) expressed in abnormal firing rates and patterns (Magnin et al., 2000; Remple et al., 2011). Even though these pathological changes are attenuated by dopaminergic medication (Brown et al., 2001; Heimer et al., 2006; Levy et al., 2002), which was taken as usual, and patients were able to perform the task well, neuronal dynamics may still have differed from those of healthy subjects with intact basal ganglia circuits.”

*D) The variability between patients should be addressed and clarified.*

We have added to the Discussion: “Inter-individual variability of the peak frequency and strength of the STN gamma increase may be related to differences in disease progression, individual stopping speed or electrode placement and type.”

*Regarding the evaluation of the right STN and the data in general the authors should perhaps discuss the fact that 3 patients were right handed but performed the task with the left finger.*

We have added to the Results: “Three right-handed patients performed the task with the left hand and thus in those the right STN was the contralateral one. However, in the remaining six the right STN was the ipsilateral STN, and thus the lack of significant right STN increase indicates that the gamma increase was specific to the contralateral STN.”

*4) The organization of the results and especially the figures is not in line with the manuscript's main emphasis. Instead of focusing on the contralateral STN, the majority of the figures contain the cortical and ipsilateral STN results. Thus, forcing important results (and especially in their raw unprocessed form) to appear as supplementary figures or not at all. For example: instead of showing in Figure 2 the total power, successful stops and failed stop within the contralateral STN, allowing the observation of the raw phenomenon we see only a processed subset but in all brain areas. It would be useful to focus on the main topic and move the other parts (such as cortical recordings) to the supplementary.*

We have reorganized the figure and now display first the raw power in the individual conditions (last regular tap, stop-signal response, failed stops, successful stops, in Figure 2) and only then the contrasts (Figure 2). We have moved the cortical recordings to Figure 3 as we think it is important to show them in the main figures to support the discussion.

*5) The study focuses on different periods and specific time windows which vary between the Results section and figures. It is unclear whether those results are robust under a more general analysis. For example: Figure 2 presents the range -100 to 700 milliseconds but the significance is checked only for 0 to 150 (based on Figure 2—figure supplement 2?). Would testing significance on a different range change the results? The figure seems to hint so, as changes do not seem to end at 150. This is true for other figures and other ranges. The authors are encouraged to revise the figures and where necessary revise the interpretations accordingly.*

We suggest that it might be unwise to perform cluster-based permutation correction over a window that is unnecessary longer than the period of critical interest for movement inhibition as this would increase the chance of a type II error. The larger the window, the less likely a cluster will survive. With a relatively small sample size, large windows would lead to an unnecessarily conservative correction. We decided to show an extended window until 700ms after the stop signal for anybody interested in how power develops several hundred milliseconds after the stop signal, even though this can only be related to evaluation rather than stopping per se.

We have now extended the window by adding frequencies between 50-60 Hz to show that the gamma differences do not extend down to 50 Hz.

We have also checked how far we can extend the time window such that the cluster remains significant. For the gamma increase after the stop signal relative to regular tapping, we could increase it to 200ms (or even to 550ms when only successful stops are considered), for the difference between successful and failed stops we could increase it to 400ms.

For the movement-related gamma increase during regular tapping, we chose the same test-window size as for the comparisons of post-stop signal gamma to keep it comparable. If we would extend the test-window to 300ms or longer, the clusters would not survive. In the legend we acknowledge that the movement-related gamma increase was relatively small:

“As power was normalized by the average power of one full tap cycle including movement, the effects were relatively small and would not survive multiple-comparison correction over the full time-window. However, movement-related β decrease and gamma increase relative to a pre-movement baseline has been repeatedly reported before (Tan, 2013; Androulidakis, 2007).”

And we also point out in the text:

“The movement-related peak was broader and weaker than the stop-related increase that peaked around 70 Hz (Figure 2—figure supplement 3).”

*6) In at least two locations in the Results section, non-significant findings are mentioned as "…did not survive multiple comparison correction over multiple time points due to the small sample size" and continue with a "…but…". This is not acceptable in statistics; something is either significant or not and cannot be forced into a "maybe" state. It should be avoided and simply stated that those results were not significant.*

We have removed the respective statements. Our intention had been to point out that the differences in power at the peak seemed to be relatively consistent across subjects, but the sample size was too small in the reduced subset for it to survive multiple comparison correction.